# Thematic-LM: a LLM-based Multi-agent System for Large-scale Thematic Analysis

## Abstract

Thematic analysis (TA) is a widely used qualitative method for identifying underlying meanings within unstructured text. However, TA requires manual processes, which become increasingly labour-intensive and time-consuming as datasets grow. While large language models (LLMs) have been introduced to assist with TA on small-scale datasets, three key limitations hinder their effectiveness on larger datasets. First, current approaches often depend on interactions between an LLM agent and a human coder, a process that becomes challenging with larger datasets. Second, with feedback from the human coder, the LLM tends to mirror the human coder, which provides a narrower viewpoint of the data. Third, existing methods follow a sequential process, where codes are generated for individual samples without recalling or adapting previous codes and associated data, reducing the ability to analyse data holistically. To address these limitations, we propose Thematic-LM, an LLM-based multi-agent system for large-scale computational thematic analysis. Thematic-LM assigns specialised tasks to each agent, such as coding, aggregating codes, and maintaining and updating the codebook. We assign coder agents different identity perspectives to simulate the subjective nature of TA, fostering a more diverse interpretation of the data. We applied Thematic-LM to the Dreaddit dataset and the Reddit climate change dataset to analyse themes related to social media stress and online opinions on climate change. We evaluate the resulting themes based on trustworthiness principles in qualitative research. Our study reveals significant insights, such as assigning different identities to coder agents promotes divergence in codes and themes.

## Keywords

Computational Social Science, Thematic Analysis, Large Language Model, Multi-agent System

**ACM Reference Format:**
Anonymous Author(s). 2024. Thematic-LM: a LLM-based Multi-agent System for Large-scale Thematic Analysis. In *Proceedings of* . ACM, New York, NY, USA, 10 pages. https://doi.org/XXXXXXX.XXXXXXX

## 1 Introduction

The growing availability of unstructured text data, particularly from social media, presents both an opportunity and a challenge for researchers [19]. While such data can hold valuable insights,

**Figure 1: Differences between topic modelling, sentiment analysis and thematic analysis, illustrated through examples of climate change-related posts.**

analysing them effectively requires robust methods to extract meaning from vast volumes of information. Computational approaches, such as topic modelling [1] and sentiment analysis [8], are designed to handle large datasets but often produce descriptive results. These methods capture surface-level patterns but fail to uncover the deeper, context-specific meanings within the data. In contrast, qualitative methods, such as thematic analysis (TA) [5], are designed to explore nuanced interpretations by focusing on the subjective experiences and contexts that shape the data. As shown in Fig. 1, a single topic may encompass underlying themes that provide richer, more meaningful interpretations. However, TA is labour-intensive and time-consuming, requiring manual processes [6] such as familiarization, coding, theme development, and interpretation, which become increasingly burdensome as the dataset grows. For larger datasets, a team of trained coders is often required to work collaboratively to manage the volume while maintaining the reliability and credibility of the results [46], which is both expensive and logistically challenging. The challenge remains in combining the scalability of computational methods with the depth of qualitative approaches.

Recent advances in large language models (LLMs) have opened new possibilities for automating thematic analysis, as LLMs have demonstrated impressive capabilities in processing unstructured data by learning from vast corpora of texts [2, 26, 48]. Researchers have begun applying LLMs as single agents to assist in the thematic analysis process [12, 13, 16, 25]. However, existing approaches have three major limitations. First, current computational thematic analysis methods require interaction with a human coder. The human coder needs to be familiar with the entire dataset, oversee the LLMs' outputs, and provide feedback to the LLMs, which is infeasible given a large dataset. Second, due to the iterative feedback process, the LLMs often tend to imitate the human coder's perspective, producing results that mirror the coder's viewpoint [12], limiting the diversity of viewpoints and resulting in a narrower analysis [12]. Third, current approaches to thematic analysis are sequential; LLMs do not revisit previously coded data to update codes as new information arises, undermining a key principle of thematic analysis: ensuring that codes accurately reflect consistent meanings across

the dataset. Apart from the limitations, evaluating the themes generated by LLMs presents further challenges due to the volume of data and the subjective nature of thematic analysis. While qualitative evaluation by humans becomes impractical at scale, automatic evaluation metrics, such as inter-rater reliability [35], often assess the similarity between the LLMs' output and manual results and overlook the fact that thematic analysis can reflect multiple valid perspectives within the data. Consequently, lower inter-rater reliability may not necessarily indicate lower-quality thematic analysis.

To address these limitations, we introduce an LLM-based multi-agent system for large-scale computational TA, which we term Thematic-LM. Thematic-LM assigns distinct components of TA as specialised tasks to individual LLM agents, fully automating the process. To simulate the process of refining codes in response to new data, we implement an adaptive codebook that stores prior codes and their corresponding quotes. A reviewer agent retrieves similar codes and quotes from the codebook, compares them to the new data, and updates the codes accordingly. We allocate coder agents with different identity perspectives to generate different views on the codes and themes. Additionally, we analyse the quality of themes based on the computational adaptation of the principles of trustworthiness in qualitative research [27, 29, 40], including credibility, dependability, confirmability, and transferability. Our main contributions are as follows:

- We propose **Thematic-LM**, an LLM-based multi-agent system for large-scale computational thematic analysis. To the best of our knowledge, Thematic-LM is the first to employ multiple LLM agents for qualitative analysis.
- We encourage different perspectives within the themes by assigning different identities to the LLM agents, prompting them to reflect on their identities while performing the analysis.
- We apply Thematic-LM to the Dreaddit [49] and the Reddit climate change dataset [1], uncovering underlying themes regarding social media stress and public opinions on climate change.

Our key findings are as follows:

- Themes from Dreaddit shows that stress-related posts reflect various levels of human needs resonating with Maslow's Hierarchy of Needs [38], from physiological needs to self-actualisation.
- Themes from the Reddit climate change dataset indicate that online opinions regarding climate change are multidimensional, capturing psychological, societal, and systemic aspects, such as eco-anxiety [43] and generational divides in perspectives.
- We found that assigning the same identities to coder agents within Thematic-LM resulted in high inter-rater reliability, whereas assigning different identities to coder agents within Thematic-LM reduces inter-rater reliability but fosters a broader and more diverse understanding of the data.

---

[1] https://www.kaggle.com/datasets/pavellexyr/the-reddit-climate-change-dataset

## 2 Related Work

**Social Media Data Analytic.** With over a billion people using social media, enormous amounts of unstructured data are generated through daily interactions on these platforms [19]. Various machine learning techniques have been employed to extract insights to handle the scale of such data. Topic modelling approaches [4, 11, 57] are applied to uncover abstract topics or clusters of similar content from large datasets. Sentiment analysis [8, 39, 54] focuses on determining the emotional tone or attitude behind a piece of text, such as classifying whether social media posts reflect positive, negative, or neutral sentiment. Other classification approaches [45, 47] are typically employed to categorize social media posts into predefined categories, such as news, entertainment and sports. However, these methods tend to produce high-level categorizations and descriptive outputs, offering surface-level insights into the data, which do not capture deeper, contextual meanings or allow for nuanced interpretations of complex social media interactions. Our Thematic-LM automates TA through a multi-agent system with multiple coders, enabling deeper exploration of underlying meanings and perspectives from large-scale datasets.

**Computational Thematic Analysis.** Several studies have explored the use of LLMs for automating TA. De Paoli [13] and Drápal et al. [16] applied LLMs to relatively small datasets by guiding the models through structured, step-by-step coding instructions. Drápal et al. [16] found that LLM performance closely aligns with human coders when iterative feedback is provided. Similarly, Dai et al. [12] proposed a feedback loop where expert input helps refine the LLM's output. While these approaches demonstrate promise, their reliance on human intervention and focus on small datasets limit their scalability. In contrast, Thematic-LM assigns a team of LLM agents to handle different components of TA, fostering a broader perspective by simulating independent coders. Each coder agent is given a unique identity, encouraging analysis from diverse viewpoints. Additionally, Thematic-LM employs an adaptive codebook that revisits and updates previously coded data, ensuring scalability and adaptability to large datasets.

**LLM-based Multi-agent System.** Recent research has shown that collaboration between multiple LLM agents can enhance inter-consistency [53], improve factuality and reasoning [17], and encourage divergent thinking [31]. Motivated by the benefits, various LLM-based multi-agent systems have been developed [10, 30, 52]. Multi-agents are often employed for problem-solving or simulation. For example, Hong et al. [21] uses specialised LLM agents as a software engineering team for developing applications collaboratively. Chan et al. [9] proposed ChatEval, which uses multi-agent debate to evaluate the quality of LLM outputs. For research on simulation, Zhang et al. [55] explored simulating collaborative intelligence in human society by assigning LLM agents various personal traits and thinking styles. Similarly, Park et al. [42] established a community of 25 agents in a sandbox environment simulating a small town, while Kovač et al. [28] constructed a school environment with LLM agents to explore developmental psychology. Moreover, Zhao et al. [56] examine the competition between LLM agents by simulating a virtual town with restaurant agents competing over customer agents. Thematic-LM focuses on TA as a problem-solving task and

**Figure 2: Thematic-LM consists of coder, aggregator, reviewer, and theme coder agents, organized into two stages: coding and theme development. In the coding stage, multiple coder agents independently analyse text data and output codes and corresponding quotes to the code aggregator. The code aggregator refines and organizes the codes before sending them to the reviewer. The reviewer maintains an adaptive codebook, ensuring updated codes are consistent with prior data. In the theme development stage, theme coder agents use the codebook to identify themes, which are then refined by the theme aggregator to produce the final themes.**

simulates coders with different identities to encourage a broader viewpoint regarding the data.

## 3 LLM-based Multi-agent System for Thematic Analysis

We adopt the inductive thematic analysis (TA) approach outlined by Braun and Clarke [5]. In TA, coding identifies items of analytic interest from the data and assigns short-phrase labels, while themes are built by synthesizing and refining insights from these codes. In traditional team-based TA, each coder often works independently to generate codes, followed by regular meetings to compare and consolidate codes, reducing redundancy or overlap and ensuring cohesion in the analysis [46]. Inductive TA is data-driven, which develops themes from the data rather than with a predefined codebook of themes [6]. In contrast to the conventional predefined codebook approach, we implement an adaptive codebook that continuously updates codes throughout the coding process, accommodating new data and insights. Building on previous work of computational TA [12, 13, 16], Thematic-LM performs TA in two stages: coding and theme development. In the coding stage, the codebook is finalized as codes are generated and refined, while the theme development stage focuses on synthesizing themes from the codebook. We provide details of the system in Section 3.1 and the coders' identities in Section 3.2.

### 3.1 Multi-agent System

As illustrated in Fig. 2, our multi-agent system consists of three types of LLM agents: coder, aggregator, and reviewer. Each agent has a specialised role in the TA process, contributing to coding and theme development stages to fully automate these tasks. The agents are implemented with conversational agents from AutoGen [52].

**Coder Agents** are responsible for coding in the first stage and identifying themes in the second stage. In the coding stage, the coders are instructed to write one to three codes for each piece of

data to capture concepts or ideas with the most analytical interest. For each code, the coder extracts a representative quote from the data as evidence. The resulting codes, quotes and corresponding quote IDs are passed to the code aggregator agent. During the theme development stage, the coders are given a complete version of the codebook from the coding stage. The codebook is compressed with LLMLingua [23, 24] to reduce token costs. The coder agents then analyse the codes and associated quotes holistically to identify overarching themes that reflect deeper insights into the data. These themes, along with theme descriptions and the most relevant quotes, are then passed to the theme aggregator.

**Aggregator Agents** refine and organize the outputs from the coder agents into structured formats suitable for the next stage. During the coding stage, the code aggregator merges codes with similar meanings, retaining differences where necessary, and organizes the codes, quotes, and quote IDs into a JSON format, which the reviewer agent uses to update the codebook. Similarly, in the theme development stage, the theme aggregator refines and organizes the identified themes and associated quotes, merging similar themes and outputting the final themes in JSON format.

**Reviewer Agent** operates exclusively during the coding stage, maintaining and updating the codebook. This codebook stores previous codes, their corresponding quotes, and quote IDs in JSON format. Codes are represented both as texts and as embeddings, generated using a Sentence Transformer model [44]. The reviewer agent processes new codes and quotes from the aggregator and retrieves the top-$k$ similar codes and quotes from the codebook by computing the cosine similarity between their embeddings. The reviewer compares the new codes and quotes with existing codes and quotes to determine whether the new codes can be updated based on prior information and whether similar existing codes can be merged. After making these decisions, the reviewer updates the codebook to save new codes and quotes and merge previous

codes into new ones. The reviewing and updating process is crucial in TA, as it plays a central role in ensuring the codes remain dynamic, interpretative, and responsive to the data. Once finalized, the codebook is passed to the theme development stage.

**Evaluation** We evaluate the quality of themes based on the principles of trustworthiness in qualitative research [27, 29, 40]. Existing metrics used in computational TA, such as inter-rater reliability [35], assume that there is one "correct" set of themes to match against for measuring the level of accuracy. We propose that trustworthiness principles, which emphasize meaningful, coherent, and data-grounded analysis, provide a more robust framework for evaluating themes. As shown in Fig. 3, We adopt the trustworthiness principles for evaluating the computational TA approaches: (1) *Credibility and Confirmability*: Credibility evaluates whether the themes accurately represent the data, while confirmability assesses whether the themes are data-driven rather than driven by biases. We measure credibility and confirmability at the same time by retrieving the associated data through quote IDs and assigning an evaluator agent to compute the percentage of data that the quote and themes are consistent with. The inconsistency with the data can be caused by hallucinations or internal biases within the LLM models. (2) *Dependability*: asses whether the same process can be repeated by a separate researcher and reveal similar findings. The dependability of the computational approach can be measured by repeating the process and measuring the inter-rater reliability of the resulting themes. We measure the inter-rater reliability in themes by conducting the TA several times and computing the average pairwise ROGUE scores [32], which measures the amount of overlap between the themes. For each pair of theme sets $A$ and $B$, we first calculate the ROUGE-1 and ROUGE-2 scores by using sets $A$ as the reference set:

$$\begin{aligned} \text{ROUGE-1}_{A \to B} &= \frac{\text{Number of overlapping unigrams in } B}{\text{Total number of unigrams in } A} \\ \text{ROUGE-2}_{A \to B} &= \frac{\text{Number of overlapping bigrams in } B}{\text{Total number of bigrams in } A} \end{aligned} \quad (1)$$

We calculate the ROUGE-1 and ROUGE-2 scores by using sets $B$ as the reference set and compute the average of the ROUGE-1 and ROUGE-2 scores for the pair of sets:

$$\begin{aligned} \text{ROUGE-1} &= \frac{1}{2} \left( \text{ROUGE-1}_{A \to B} + \text{ROUGE-1}_{B \to A} \right) \\ \text{ROUGE-2} &= \frac{1}{2} \left( \text{ROUGE-2}_{A \to B} + \text{ROUGE-2}_{B \to A} \right) \quad (2) \\ \text{ROUGE} &= \frac{1}{2} \left( \text{ROUGE-1} + \text{ROUGE-2} \right) \end{aligned}$$

(3) *Transferability* assesses whether the identified themes and codes can be meaningfully applied to other contexts or datasets with similar characteristics. We measure the transferability by splitting the dataset into a training and validation set, where we perform TA separately and measure whether the themes from the training set can transfer to the themes in the test set by computing the overlap between themes via pairwise ROGUE scores shown in Eq. (2).

## 3.2 Coder Identities

TA inherently embraces subjectivity, recognizing that researchers bring their own perspectives, assumptions, and interpretations to the data [6, 18]. The identification of themes is guided by the

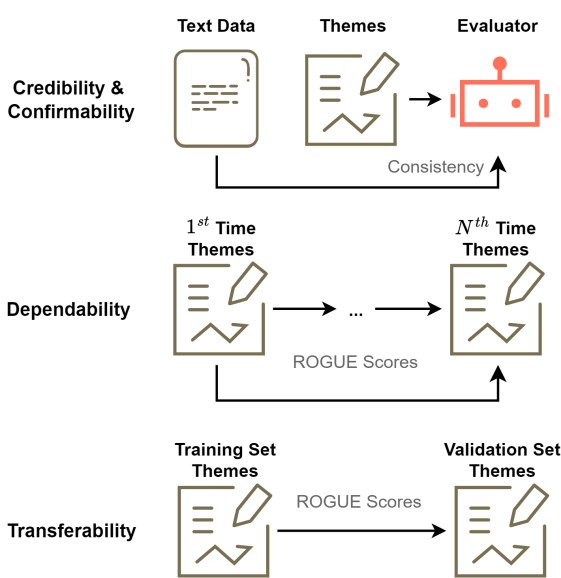

**Figure 3: Evaluation framework: we employ an evaluator agent to check the consistency between the themes and associated data to assess the credibility and confirmability of the themes. To assess their dependability, we repeat the thematic analysis (TA) process $N$ times and examine whether the themes remain stable by computing the overlap between themes. For transferability between similar datasets, we perform TA independently on two split data sets and compute the overlap between themes.**

coder's insights and understanding, which plays an active role in deciding what is meaningful in the data. Consequently, the same data may yield different themes depending on who is conducting the analysis. Coders may interpret the same information in diverse ways, especially when they come from varied social, cultural, or professional backgrounds. This variability does not undermine the reliability of the analysis but instead highlights the subjectivity that enriches qualitative research. The subjective nature of thematic analysis allows it to delve deeply into human experiences, emotions, and meanings while providing the contextual understanding needed to explore nuanced social and cultural issues [18, 40].

In previous work on computational thematic analysis, the LLM's outputs are aligned with a human coder through iterative feedback. In contrast, Thematic-LM simulates coders with varied backgrounds to foster diverse perspectives in data interpretation. In profiling the coder agents, we draw from existing literature on different viewpoints and opinions related to the subject matter, assigning distinct identities to each agent. These agents are instructed to interpret the data through the lens of their assigned identities, reflecting on how someone with such a background might perceive and analyse the information. This approach allows us to explore the diversity of perspectives that may emerge from the data and offers a way to measure the divergence between coders' interpretations due to different backgrounds.

**Table 1: Themes and description of the themes produced by Thematic-LM on the Dreaddit dataset, compared with topic modelling and sentiment analysis results.**

| Theme | Description | Related Topic | Sentiment Score |
|-------|-------------|---------------|-----------------|
| Navigating Emotional and Mental Health Challenges | The experiences of individuals dealing with anxiety, mental health concerns, and the impact of emotional stressors on daily life. | therapist_to_therapy | 0.00 |
| Impact of Economic Stress and Resource Scarcity | The financial struggles and efforts to manage limited resources and economic instability. | homeless_the_to_have | 0.10 |
| Familial Dynamics and Responsibilities | The complex relationships within families, the burden of roles, and the tension between obligations and emotional needs. | food_you_for | 0.15 |
| Coping with Academic and Professional Pressures | The stress associated with academic performance and the pressures to succeed in professional life. | job_and_have | 0.20 |
| Interpersonal Conflict and Relationship Stress | The conflicts in personal relationships, whether with romantic partners, friends, or colleagues, the emotional and mental strain caused by unresolved tensions or disagreements. | sex_that_sexual | 0.15 |
| Seeking Validation and Emotional Support | The frequent attempts by individuals to gain reassurance or validation from others, often by sharing their struggles and emotions openly in the hope of receiving empathy or encouragement. | you_are_support | 0.35 |
| The Dichotomy of Online and Offline Identities | The differences between one's real and virtual personas and the impact of social media on identity and interactions. | he_him_me | 0.25 |

## 4 Experiments and Analysis

We conduct a series of experiments to assess Thematic-LM's effectiveness in performing thematic analysis on social media datasets. At the time of writing, no other multi-agent systems were designed for qualitative analysis, such as thematic analysis. Therefore, our experiments focus primarily on Thematic-LM itself. Specifically, we aim to answer the following questions:

- What insights can Thematic-LM uncover from the Dreaddit and the Reddit climate change dataset? (Section 4.1)
- Does Thematic-LM produce higher quality themes compared to a single LLM agent? (Section 4.2)
- How do the different coder identities influence the codes and themes? (Section 4.3)

**Experimental Setup**. We use GPT-4o to serve as the LLM agent accessible through the OpenAI API *gpt-4o-2024-05-13*[2]. The *temperature* and *top_p* are set at the default value of one. For each code and theme, the agents save up to 20 of the most relevant quotes associated with the concept. The reviewer retrieves the top 10 most similar codes for each new code. To measure the dependability, we conducted the TA three times and calculated the average pairwise ROGUE scores between the resulting sets of themes. To measure transferability, we split the dataset into a 50% training set and a 50% validation set. We employ Thematic-LM to analyse the Dreaddit [49] and the Reddit climate change dataset. Dreaddit contains over 190k posts from subreddits related to abuse, anxiety, financial issues, PTSD and relationship problems. The Reddit climate change dataset consists of 4.6 million Reddit posts and comments that mention the terms "climate" and "change". The source code is available in the supplementary material.

----

[2]https://platform.openai.com/docs/models/gpt-4o

## 4.1 Thematic Analysis of Social Media Data

We assign two coder agents and two theme coder agents for TA on Dreaddit and the Reddit climate change dataset. The agents are provided with the instructions only, with no personal identities given to the agent. To compare the TA results, we perform topic modelling and sentiment analysis on the datasets. We employ BERTopic [20] to categorize the posts into topics and RoBERTa [33] from TweetNLP [7] for sentiment analysis. We set the number of neighbours, number of components for UMAP [37] and minimum cluster size for HDBSCAN [36] in BERTopic as 15, 10 and 10, respectively. We use cosine similarity as the distance metric in HDBSCAN. For each theme, we select the most relevant topic by looking at the majority of the topics of the data points associated with the theme through the quote IDs. Similarly, we computed the average sentiment of the data points. Sentiment scores of zero, one and two denote negative, neutral and positive, respectively.

**Dreaddit datset**. As shown in Table 1, Thematic-LM produced seven themes from the Dreaddit dataset. The sentiment analysis returns mostly negative labels for the data associated with the themes. We observe that the themes identified by Thematic-LM on the Dreaddit dataset highlight a broader and more meaningful understanding of the data compared to the related topics generated by topic modelling. For instance, the theme "Navigating Emotional and Mental Health Challenges" captures the users' struggles with anxiety and emotional stressors. In contrast, the related topic "therapist_to_therapy" provides a more fragmented association of words, missing the depth in the narrative. Similarly, "Impact of Economic Stress and Resource Scarcity" encapsulates the daily struggles of managing limited resources, while the related topic "homeless_the_to_have" only loosely connects words around homelessness and possession, failing to capture the specific challenges

**Table 2: Themes and description of the themes produced by Thematic-LM on the Reddit climate change dataset, compared with topic modelling and sentiment analysis results.**

| Theme | Description | Related Topic | Sentiment Score |
|---|---|---|---|
| Emotional Burden of Climate Change Awareness | The complex emotions individuals face, such as anxiety, guilt, frustration, and helplessness, stemming from the overwhelming nature of climate change and a perceived lack of control over its outcomes. | climate_change_years | 0.20 |
| Generational and Cultural Disconnection | The perceived gaps in understanding and values across different generations and cultures, which are often exacerbated by rapid societal and technological changes. | denier_climate_scope_change | 0.30 |
| Call for Collective Action and Unity | Emphasizing the necessity for collective action and unity in addressing societal challenges, including political divisions and significant issues like climate change and economic inequality. | jobs_people_us | 1.30 |
| Critique and Skepticism of Political and Economic Systems | A critical examination of current political and economic systems, highlighting concerns about inequality, inefficiency, and the shortcomings of existing policies. | bank_companies_billion | 0.25 |
| Personal and Community Resilience | Personal and communal efforts to adapt to the impacts of climate change, emphasizing the importance of strengthening social connections and local initiatives | people_change | 0.85 |
| Role of Technology in Climate Solutions | The potential of technology and innovation to mitigate climate change effects, including renewable energy advancements, carbon capture technologies, and sustainable agriculture practices. | energy_nuclear_power | 0.90 |
| Impact on Biodiversity and Ecosystems | Individuals express worries about endangered species, habitat destruction, and the overall health of the planet's ecosystems. | meat_animals | 0.10 |
| Climate Migration and Displacement | The challenges faced by communities and individuals who are forced to relocate due to climate change impacts such as rising sea levels, extreme weather events, and resource scarcity. | housing_cities_city | 0.20 |

**Table 3: Comparison of the theme quality scores on the Dreaddit dataset.**

| Method | Credibility & Confirmability | Dependability | Transferability |
|---|---|---|---|
| Single | 0.63 | 0.45 | 0.41 |
| Single (Codebook) | 0.75 | 0.61 | 0.67 |
| System (1 Coder) | 0.92 | **0.81** | 0.86 |
| System (2 Coders) | **0.94** | 0.78 | **0.87** |

**Table 4: Comparison of the theme quality scores on the Reddit climate change dataset.**

| Method | Credibility & Confirmability | Dependability | Transferability |
|---|---|---|---|
| Single | 0.66 | 0.56 | 0.73 |
| Single (Codebook) | 0.74 | 0.69 | 0.78 |
| System (1 Coder) | 0.96 | 0.84 | **0.90** |
| System (2 Coders) | **0.98** | **0.86** | 0.89 |

individuals face in their economic lives. This comparison illustrates how thematic analysis delves into the underlying meanings and human experiences, offering a much more insightful and comprehensive picture than topic modelling, which often yields superficial groupings of co-occurring terms.

The themes reflect various levels of human needs, resonating with Maslow's Hierarchy of Needs [38]. "Navigating Emotional and Mental Health Challenges" and "Impact of Economic Stress and Resource Scarcity" correspond to Maslow's foundational physiological and safety needs, as they involve mental well-being and financial

stability. "Familial Dynamics and Responsibilities" and "Interpersonal Conflict and Relationship Stress" align with belongingness and love needs, highlighting the importance of relationships and emotional bonds in individuals' lives. Meanwhile, "Coping with Academic and Professional Pressures" and "Seeking Validation and Emotional Support" relate to esteem needs, where individuals seek recognition, achievement, and emotional validation. Finally, "The Dichotomy of Online and Offline Identities" reflects the higher-order need for self-actualization as individuals navigate personal identity and the complexities of presenting themselves in digital and real-world environments.

**The Reddit climate change dataset**. As shown in Table 2, the eight themes identified by Thematic-LM present a more nuanced and interconnected understanding of climate change discourse, emphasizing emotional and social dimensions rather than solely categorizing discussions by co-occurring words. The emotion captured by the themes generally aligns with the sentiment score. The "Emotional Burden of Climate Change Awareness" theme highlights the psychological distress, anxiety, and feelings of helplessness that individuals face, reflecting the concept of eco-anxiety [43]. This emotional struggle is intertwined with the "Generational and Cultural Disconnection", which points to the gaps in understanding and values that can arise between different generations, further complicating collective responses to climate change. The theme "Call for Collective Action and Unity" underscores the necessity for collaboration in addressing climate-related challenges, emphasizing a shared responsibility that can help foster social cohesion. This is

complemented by the theme "Personal and Community Resilience", which showcases the importance of local initiatives and social connections as individuals and communities adapt to the changing environment. Meanwhile, the "Critique and Skepticism of Political and Economic Systems" reflects a growing awareness of systemic failures and the need for significant reforms to ensure effective climate action. The theme "Role of Technology in Climate Solutions" highlights the potential for innovation and advancements in technology to mitigate the impacts of climate change, showcasing a hopeful perspective amid the challenges. Finally, the theme "Impact on Biodiversity and Ecosystems" serves as a reminder of the broader ecological implications of climate change, emphasizing the interconnectedness of human actions and environmental health. Together, these themes illustrate a complex multifacet of emotional, social, and systemic factors shaping climate change discourse from social media.

## 4.2 Quality of Themes

To investigate the benefits brought by the adaptive codebook and the multi-agent system, we compare the quality of themes between a single LLM agent, a single LLM agent with an adaptive codebook, Thematic-LM with one coder for coding and theme development and Thematic-LM with two coders for coding and theme development. The single LLM agent is instructed to first label the data sequentially with codes, define themes from the codes, and save the data IDs of the most relevant codes for each theme. The single LLM agent with the adaptive codebook adds steps for retrieving similar codes for comparison and saving codes and quotes into the codebook. After the coding stage, the single LLM agent takes the codebook as input and defines themes from the codebook. The coder agents are given instructions without assigning any identities.

As shown in Tables 3 and 4, we observe that the introduction of an adaptive codebook improves the quality of the themes of the LLM agent, and multi-agents perform better than single agents. The credibility & confirmability score of the single agent with the codebook is improved due to the LLM agent having access to retrieve past quotes, which provides a chance to reflect on past codes and quotes whenever similar data arrive. The similar codes and associated quotes together give a more holistic view and improve the context understanding of the data, which makes the coding less affected by randomness brought by a single data sample. The improvement in both the dependability and transferability of the themes shows this. The multi-agent systems have higher credibility & confirmability, dependability and transferability scores. In the multi-agent system, the distribution of specialised tasks has made the tasks simpler and shorter for each agent, improving factuality and reducing hallucination brought by doing complex tasks. This has led to more stable and transferable themes across different runs.

## 4.3 Divergence of Pespectives

We aim to investigate whether assigning different identities can broaden the views of the TA and the effects of assigning identities to coders. We conducted the experiments on the Reddit climate change dataset, as climate change is a polarizing issue due to the intersection of social, economic, political and cultural values, which lead to divergent opinions [14, 15]. The subjectivity of TA and the scale

**Table 5: Examples of themes not captured in Thematic-LM with no assigned coder identities but emerged in Thematic-LM where coders are assigned different identity perspectives.**

| Theme | Description |
|---|---|
| Economic Impact of Climate Policies | Concerns about the economic consequences of aggressive climate regulations, particularly their impact on industries and job markets. |
| Environmental Stewardship | The deep responsibility to protect and maintain the natural environment, viewing humans as caretakers of the earth. |
| Scepticism of Climate Science | Questioning about the extent of human influence on global warming and discuss whether climate change is part of the natural cycle. |
| Environmental Justice and Vulnerable Communities | Highlights the disproportionate impact of climate change on marginalized communities, advocating for policies that address environmental justice and protect vulnerable populations. |

of the dataset might lead to some views being underrepresented in the resulting themes. We assign five coders with different identity perspectives: (1) *Human-Driven Climate Change Agent*: This agent adopts the widely accepted scientific view that human activities are the primary drivers of climate change [15]. It focuses on the role of industrialization, fossil fuel emissions, deforestation, and other anthropogenic activities in accelerating global warming. The agent emphasizes the need for policy reforms, renewable energy adoption, and collective global action to mitigate the impact of human-caused environmental degradation. (2) *Natural Climate Change Agent*: This agent approaches climate change from the viewpoint that it is a natural phenomenon, part of Earth's long-term climatic cycles. It reflects the arguments that climate fluctuations have occurred over millennia due to factors like solar radiation, volcanic activity, and ocean currents, suggesting that current climate shifts may not be solely due to human activities [3, 34]. This perspective is often used to critique policies perceived as overemphasizing the human impact on the environment. (3) *Progressive View Agent*: The progressive agent is given the progressive perspective rooted in environmental justice, equity, and sustainability, advocating for systemic changes that address not only environmental issues but also social inequalities exacerbated by climate impacts [15]. The agent emphasizes green technologies, grassroots activism, and policies that ensure vulnerable communities are not disproportionately affected. (4) *Conservative View Agent*: This agent reflects the conservative perspective on climate change, focusing on gradual, market-driven solutions rather than large-scale regulatory interventions [15]. It prioritizes economic stability, energy independence, and limited government involvement in climate policies. From this viewpoint, climate action should not jeopardize economic growth, jobs, or individual freedoms. (5) *Indigenous View Agent*: The Indigenous agent operates from the perspective that climate change is deeply intertwined with human relationships with nature and the environment [50, 51]. It emphasizes traditional ecological knowledge, the interconnectedness of all living beings, and the sacred responsibility to care for the land. This agent highlights climate change's

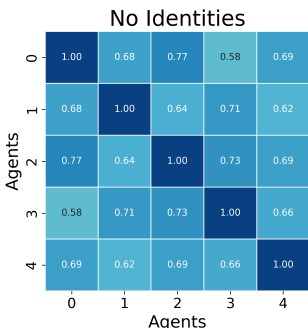 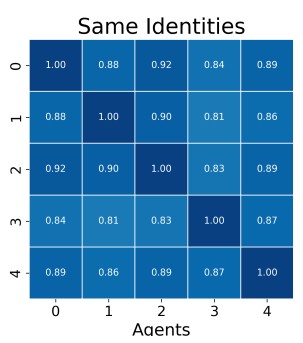 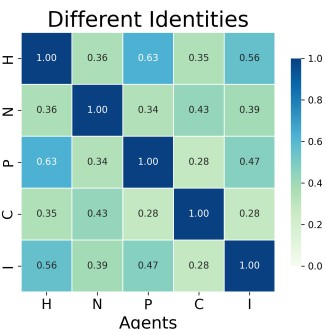

**Figure 4: Comparison of codes and themes generated by five coders with no identities given, with the same identities of "human-driven climate change" given, and with five different identities. The five different identities are agents which are instructed to believe in "human-driven climate change" and "climate change as a natural cycle" and instructed to act with "progressive view", "conservative view" and "Indigenous view", denoted as H, N, P, C, and I respectively. The pairwise ROGUE scores measure the differences between codes and themes from different agents.**

cultural, spiritual, and community-based dimensions, particularly the impacts on Indigenous lands and ways of life.

To measure the effects of identity perspectives on the coders, we measure the inter-rater reliability via pairwise ROGUE scores (Eq. (2)) among the coder agents during the coding and theme development stage within Thematic-LM. For example, during the coding stage, we compare codes between each pair of agents by calculating the ROGUE scores. The final scores for each system are the average scores from the coding and theme development stages. We calculate the ROGUE scores as the average between ROGUE-1 and ROGUE-2 scores. We compare the differences between assigning no identities to coder agents, assigning the same identities to coder agents and assigning different identities to coder agents. For the first system, we assign five coders for coding and theme development without any identities. Second, we assign the five coders the same identities of the "human-driven climate change" view to measure the effect of having the same identity perspectives. Third, we assign the five different identities to the coders to measure the divergence of perspectives. As shown in Fig. 4, the agents with different identities produced divergent codes and themes with overall lower ROGUE scores than agents with no identities assigned, while the agents with the same identities produced more similar codes than agents with no identities assigned. For agents with no identities assigned, there are some variations in the codes and themes, as the ROGUE scores indicate there are about 58% to 77% of overlap words and word pairs. While the codes and themes of the agents with different identities diverge from each other, there is a relatively higher overlap of codes and themes for agents with related views. For example, there is some overlap between codes from human-driven climate change and progressive views, such as codes related to collective action to reverse the effect of pollution on climate change. The results from the assignment of identity perspectives mirror how the views of the coders might affect their interpretation of the data in TA. The convergence and divergence of codes and themes simulate and resonate with social identity theory [22] and confirmation bias [41]. Social identity theory emphasizes that the categorization of individuals into social groups influences their attitudes, behaviours, and

interpretation of information, while confirmation bias is the tendency to search for, interpret, and favour information that confirms one's pre-existing beliefs or hypotheses.

Although in the Thematic-LM with five different coders, the codes and themes diverge from each other, we have instructed the code and theme aggregator to retain the different codes and themes to maintain the different perspectives. As a result, the resulting themes are more diverse than Thematic-LM with no agent identities assigned. For Thematic-LM with five different agent identities, 15 themes are identified from the Reddit climate change dataset. As shown in Table 5, we illustrate examples of themes that are not captured in Thematic-LM with no given coder identities but have emerged in Thematic-LM with different coder identities. We observe that with different identity perspectives, the agents might highlight unaddressed issues by considering different viewpoints. For example, themes such as "Economic Impact of Climate Policies" and "Scepticism of Climate Science" reflect concerns and beliefs that differ from those captured in a more homogenized analysis without different identities.

## 5 Conclusion

We presented Thematic-LM, the first LLM-based multi-agent system for large-scale thematic analysis. Thematic-LM addresses key challenges in the computational thematic analysis of large-scale datasets by distributing the tasks among specialised agents and maintaining an adaptive codebook for maintaining and updating codes. We employ Thematic-LM to analyse the Dreaddit and Reddit climate change datasets. We evaluate the quality of themes based on the principles of trustworthiness in qualitative research. Furthermore, we experimented with assigning different perspectives to the coder agent to simulate the subjective nature of thematic analysis and broaden the views captured in the themes. Our work lays a foundation for conducting qualitative research with LLM agents. Future work could investigate combining other qualitative methods and incorporating the imaging modality into the analysis.

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
