# OpenReview forum: "Thematic-LM: a LLM-based Multi-agent System for Large-scale Thematic Analysis"
_ACM.org/TheWebConf/2025/Conference — WWW 2025 Oral_

### Official Review · Reviewer_Cf24 · 2024-12-02

**Novelty:** 5
**Technical Quality:** 4

**Review:**

**Summary:**

The paper proposes an LLM multi-agent collaboration framework for qualitative thematic analysis (TA). It divides TA into sub-tasks like coding, aggregating codes, updating codebook, generating themes, and assigns different sub-tasks to different LLM agents. Moreover, it maintains an adaptive codebook to ensure consistency over large-scale datasets, and assigns coders different roles to enhance diversity of interpretation. The authors apply their methods to two datasets and conduct an in-depth analysis.

**Pros:**
1. The paper is well-organized and the writing is easy to follow.
2. The idea of leveraging LLM multi-agent collaboration for large-scale TA is innovative.
3. The experiments on two datasets are comprehensive and reveal some interesting insights.


**Cons:**
1. The presentation of methods in Section 3 is not clear enough and lacks details. For example, is each coder responsible for coding a subset of the data or the full dataset? How do the agents handle long context given the large-scale dataset? Are they equipped with modules like external memory? It would be beneficial to include some symbolic representations such as an Algorithm, and some examples of prompts and responses of different agents.
2. Some metrics are questionable. The Dependability seems to be largely affected by the randomness of LLM, such as the temperature parameter. I wonder whether this would affect the results. Moreover, the word "transferability" may be inappropriate since it usually refers to the transfer across different datasets with different domains or data distributions, while in the paper the authors just split the dataset into two subsets with similar characteristics. The meaning of this metric and its difference from other metrics should also be further clarified.
3. In experiments, the authors only compare Thematic-LM with topic modeling and sentiment analysis methods, but do not compare with other TA methods. The authors state that there are no other multi-agent systems, but I still wonder whether there are other TA methods or human analysis results for the two datasets. Without a comparison of the generated themes, it is hard to demonstrate the correctness and advantage of Thematic-LM.

**Questions:**

1. Please refer to the issues in **Cons**.
2. Assigning different identities to coder agents reduces reliability but improves diversity, so how to balance the reliability and diversity? And how to design the identities for different datasets?

**Reviewer Confidence:**

3: The reviewer is confident but not certain that the evaluation is correct

**Scope:**

4: The work is relevant to the Web and to the track, and is of broad interest to the community

---

### Official Review · Reviewer_LU6o · 2024-12-02

**Novelty:** 6
**Technical Quality:** 3

**Review:**

The authors propose Thematic-LM, an LLM-based
multi-agent system for large-scale computational thematic
analysis. Thematic-LM assigns specialized tasks to each agent, such
as coding, aggregating codes, and maintaining and updating the
codebook. They assign coder agents different identity perspectives
to simulate the subjective nature of TA, fostering a more diverse
interpretation of the data. They then apply it to the Dreaddit
dataset and the Reddit climate change dataset to analyze themes
related to social media stress and online opinions on climate change.

The study reveals some interesting findings, and for both this and the authors'
framing of the problem, I have given the paper a high novelty rating. Analysis of stress-related posts from
the Dreaddit dataset aligns with Maslow’s Hierarchy of Needs, reflecting a range
of human concerns from basic physiological needs to self-actualization.
Insights from the Reddit climate change dataset showcase the multidimensional nature
of online climate change discussions, embodying psychological aspects like
eco-anxiety and factors such as generational divides in perspectives.
Furthermore, the research illustrates that assigning identical identities to
coder agents within Thematic-LM leads to high inter-rater reliability, while
using diverse identities reduces reliability but still emphasizes broader and
more subtle interpretation of the data.

My main concern is with the technical quality of the paper, which is not at the level
of the Web conference. There are few quantitative results, and the ones that are presented
are not properly discussed and substantively fleshed out. The paper is therefore promising,
but still too preliminary for the main conference.

Other comments:

--some of the figures are unnecessary and are serving only as fillers e.g., figure 1. Similarly, Figures 2 and 3 can both be made more rigorous.

**Questions:**

None.

**Reviewer Confidence:**

3: The reviewer is confident but not certain that the evaluation is correct

**Scope:**

4: The work is relevant to the Web and to the track, and is of broad interest to the community

---

### Official Review · Reviewer_sgn8 · 2024-12-02

**Novelty:** 5
**Technical Quality:** 5

**Review:**

This paper explores the use of an LLM-based multi-agent system to conduct thematic analysis tasks. There is a body of existing research on this topic, so the novelty of this approach is in reducing or removing dependency on a human coder and any bias that the human coder may introduce and in working from a codebook. The work is generally well-written and there are few issues with the text.  The paper is realistic about its contribution and references a selection of prior work.

The authors may wish to revise the paper a little to ensure that sufficient context is given in the introduction to understand the datasets used (for example: why were the Dreadit and Reddit datasets selected as appropriate demonstrators for this approach? Line 150 states that this was done, but it would signpost the rest of the paper more effectively if the intuitions behind the selection of these datasets were foreshadowed early on). This is eventually explored in the body of the paper, but it may be more effective to introduce these themes early on.

While the authors make some very good points regarding the limitations of inter-rater reliability as a metric  - e.g. multiple valid perspectives on the data may exist, and direct inter-rater reliability is better suited to consistency and repeatability within a single frame of reference - the paper could explore the annotation question in more depth. For example, it may be helpful to discuss whether repeated runs of an agent with a specific perspective (e.g. Progressive View, Conservative View) could usefully be evaluated via essentially qualitative metrics. Additionally, it may be helpful to briefly discuss alternative metrics that do take rater perspective into account: for example, exploring whether humans who self-identify as holding a particular perspective take the view that the agent's perspective is compatible with their own. In conclusion, a question that I think would be very helpful to explore in the segment of this paper that deals with annotation is: while it is entirely reasonable to state that low inter-rater reliability does not necessarily indicate low quality outcomes, are there other approaches to scoring that might be of relevance?

The approach taken to validation in this case is largely automated, followed by an evaluation by inspection, carried out with reference to Maslow's hierarchy of needs. For validation purposes, some form of manual review (for example, manual scoring of agents' random samples for apparent validity, given the framing provided) may be worthwhile.

The discussion of thematic analysis is informative and makes good use of existing literature. Around line 354, it would be helpful to provide a brief introductory statement regarding the trustworthiness principles and their applications before moving on to their adoption, as many readers will be unfamiliar with the topic.

**Questions:**

Question: You state that Thematic-LM simulates coders with varied backgrounds to foster diverse perspectives in data interpretation. In profiling the coder agents, we draw from existing literature on different viewpoints and opinions related to the subject matter, assigning distinct identities to each agent.
For me, the unanswered question behind evaluation of this paper is: how 'true-to-life' is this process of simulation? From section 4.3, it is clear that the agents differ in their responses according to identity perspective, so that is helpful in itself. It is likely that there is some benefit to be derived from this process regardless of the fidelity of the agent to the archetype. But I am certainly curious as to the extent to which an agent is able to reflect these viewpoints, and to what extent the fidelity of this simulation is seen to matter in each context. I give as an example the Indigenous perspective: to claim that a device is capable of emulating such a perspective has potential evident ethical risks, notably that of silencing the voices of the Indigenous community in favour of a device that may not have sufficient information to speak as those voices might, and so I think it would be very interesting to explore questions around the use of such systems with members of the community in question, perhaps as part of further work.

**Reviewer Confidence:**

3: The reviewer is confident but not certain that the evaluation is correct

**Scope:**

4: The work is relevant to the Web and to the track, and is of broad interest to the community

---

### Official Review · Reviewer_3Dxs · 2024-12-03

**Novelty:** 4
**Technical Quality:** 3

**Review:**

The submission introduces an LLM-based system for thematic analysis. The idea of multi-agent systems that scale to large-scale data is enticing, and could be very valuable, but unfortunately the manuscript is vague and lacks detail and basic explanations/definitions, so the method is impossible to reproduce. Furthermore, it lacks any form of human evaluation or feedback, so the quality of output is unclear. Overall, the manuscript in its current form isn't suitable for publication at the conference.

Core concepts are unclear. For instance, it's unclear what the overlap between "theme sets" is in the computation of dependability. Reader is forced to infer these concepts from sentences like this once: "For each theme, we select the most relevant topic by looking at the majority of the topics of the data points associated with the theme through the quote ID". Also, there seems to be a significant level of overlap between the concepts of dependability and transferability. The authors point out that "the transferability assesses whether the identified themes and codes can be meaningfully applied to other contexts", but then instead of ensuring "other contexts", they just subsample from the same dataset as the one used to develop the codebooks. This results in a measure that's almost the same as the measure of dependability, the only difference being that it's based on a subsample without any averaging. Then, it's not a surprise that results in Tables 3 and 4 aren't very different for these two measures. The differences in their values may simply correspond to noise.

Also, many elements of the system are unclear. For instance, how exactly is it decided that new codes are added to the codebook? The manuscript states that the reviewer agent keeps top k similar codes, but it's not clear whether codes and quotes are embedded separately, nor how was k set and how it impacts the results. Similarly, the quality evaluation section states that the single LLM agent "adds steps for retrieving similar codes for comparison and saving codes and quotes into the codebook". How exactly? These are just a couple of examples. Unfortunately, the manuscript doesn't provide any LLM prompts, so it's not clear how these steps are performed and it's impossible to reproduce them. I'd suggest to provide key prompts in the manuscript, while all minor ones in an Appendix, keeping them well organised and clearly described. Writing lacks detail and could be better organised/structured: when details are provided then they're mixed up with high level information, e.g., the details of baselines could be provided in a separate paragraph.

Importantly, the current evaluation is performed entirely by an LLM, without any human feedback or data labelling and evaluation. I'd trust the results of this manuscript more if some sort of evaluation involving humans was applied, e.g., one of the methods used to evaluate topic models [1]. For instance, take a subset of all "the data points associated with the theme through the quote ID", then include an intruder (a data point not associated with the theme). How easily could a human find the intruder? Is the result better than for BERTopic? Also, in lines 741-742 it's mentioned that the multi-agent systems help in various ways, but the term "multi-agent" seems to put Thematic-LM with 1 coder and multiple coders in one bag, when based on Table 3 and 4 it's not clear whether the improvement due to the addition of coders is significant. Why not include more coders? Would this help or not?

Finally, while the manuscript makes comparisons with BERTtopic, it doesn't review literature on topic models, news clustering [2], or topic/cluster evaluation [1].

[1] Chang, Jonathan, et al. "Reading tea leaves: How humans interpret topic models." Advances in neural information processing systems 22 (2009).

[2] Chen, Xi, et al. "Global News Synchrony and Diversity During the Start of the COVID-19 Pandemic." Proceedings of the ACM on Web Conference 2024. 2024.


Pros:
- Results presented in Table 1 and 2 look great at face value.
- Scaling to large-scale data.

Cons:
- Vagueness about the core concepts such as "theme".
- Unclear descriptions of key elements of the system.
- It's impossible to reproduce any of these results. What were the prompts? Writing can be improved and better structured.
- Lack of any human evaluation or feedback.
- RW misses topic/clustering models and their evaluation.

**Questions:**

- What is a theme? What are theme sets?
- Does this system always start with an empty initial codebook and coders invent all codes?
- How credibility scores can be so high in Tables 3 and 4, given that there are three codes per article and probably even more per theme? Based on this, I was expecting credibility score (defined as the "% of data that the quotes and themes are consistent with") whose values are close to 1/3.

**Ethics Review Flag:**

Yes

**Reviewer Confidence:**

3: The reviewer is confident but not certain that the evaluation is correct

**Scope:**

3: The work is somewhat relevant to the Web and to the track, and is of narrow interest to a sub-community

---

### Official Review · Reviewer_zUZp · 2024-12-03

**Novelty:** 4
**Technical Quality:** 5

**Review:**

The paper proposed Thematic-LM, a multi-agent system for large-scale thematic analysis that uses coder, aggregator, and reviewer agents in their pipeline. It's adaptive codebook allows dynamic refinement of codes. The proposed system is intuitive and, in general, ThematicLM demonstrates significance through applications in mental health and climate discourse, though expanding to more datasets could strengthen the work’s credibility. Please find my detailed comments below.


**Strengths**

1. The use of multi-agent design for TA is intuitive. All three modules of the proposed pipeline (coder, aggregator and reviewer agents) is well thought and the paper addresses their roles effectively.

2. I believe the adaptive codebook is a standout engineering part of the pipeline. Its ability to revisit and refine codes as the process goes on seems like a major contributor to ThematicLM's performance.

3. The idea of assigning distinct identities to coder agents and to mimic human subjectivity is practical.

4. Evaluation based on trustworthiness is totally relevant and is better compared to the standard suite of metrics.

**Weaknesses**

1. While the method feels ‘intuitive’, I’m hesitant to call it ‘novel’. The use of multi-agent pipelines for downstream tasks is already a well-studied area in current research. That said, this alone is not the sole deciding factor for the decision (for me).

2. The structure of the paper can be improved. For example, the community working on TA is still relatively small (as the authors mention), and there are no practical baselines to compare against (as the authors mention), showing how new this line of work is. The introduction section of the paper does little to clarify the problem or explain the modules. I found it hard to understand what codes are and what the goal is till I read Section 3.1 and saw the Tables in the Experiment section.

3. In the same line of thought, Fig.1 is unable to explain the problem and the example is unclear. I suggest authors to take advantage of the appendix space and provide more suited examples to justify not just the problem statement but also justification examples that better justify the problem and explain each module in the pipeline. As a reviewer, I would have found the paper far more convincing, with better clarity from the start.

4. Since this work doesn’t rely on annotations, it would have been interesting to see the usage of other modalities from the corpus. Possible?

5. When working with sensitive mental health-related subreddits, it is suggested to have a dedicated section on ethical considerations.

6. I do understand that there are no specific baselines (as the authors mention), but that does not mean we cannot see possible comparisons between the current state-of-the-art LLMs and ThematicLM. It would have been both interesting and essential to see prompt-tuned LLMs for the same task, followed by employing your evaluation criteria.

7. The pipeline design of Thematic-LM is intuitive, but it would be interesting to see how other LLMs might contribute to the pipeline. Some analysis of how different LLMs could play distinct roles in this multi-agent setup would have added depth to the discussion.

**Questions:**

In addition to the feedback laid in weakness, here are additional questions:

1. Why were only two datasets used? I don’t have a strong opinion for or against this, but I’m curious about the rationale. Why these two specific domains: mental health and climate change, especially since they’re so diverse? The method seems like it could be applied to a wide range of topics. Would it be possible to run and benchmark Thematic-LM on a more varied set of datasets? If authors are going to diversify, it might be better to go all in and test across a broader spectrum of applications.

2. How does Thematic-LM address scenarios where thematic convergence is required (e.g., policy-making) rather than divergence?

**Reviewer Confidence:**

4: The reviewer is certain that the evaluation is correct and very familiar with the relevant literature

**Scope:**

4: The work is relevant to the Web and to the track, and is of broad interest to the community